# Heterogeneous Light Conditions Reduce the Assimilate Translocation Towards Maize Ears

**DOI:** 10.3390/plants9080987

**Published:** 2020-08-04

**Authors:** Guopeng Chen, Hong Chen, Kai Shi, Muhammad Ali Raza, George Bawa, Xin Sun, Tian Pu, Taiwen Yong, Weiguo Liu, Jiang Liu, Junbo Du, Feng Yang, Wenyu Yang, Xiaochun Wang

**Affiliations:** 1College of Agronomy, Sichuan Agricultural University, Chengdu 611130, China; 2018101009@stu.sicau.edu.cn (G.C.); chenhong1@stu.sicau.edu.cn (H.C.); shikai@stu.sicau.edu.cn (K.S.); Razaali0784@yahoo.com (M.A.R.); bawa_george@yahoo.com (G.B.); sunxin@sicau.edu.cn (X.S.); TianPu@sicau.edu.cn (T.P.); yongtaiwen@sicau.edu.cn (T.Y.); 13299@sicau.edu.cn (W.L.); jiangliu@sicau.edu.cn (J.L.); junbodu@sicau.edu.cn (J.D.); f.yang@sicau.edu.cn (F.Y.); mssiyangwy@sicau.edu.cn (W.Y.); 2Sichuan Engineering Research Center for Crop Strip Intercropping System, Key Laboratory of Crop Ecophysiology and Farming System in Southwest China (Ministry of Agriculture), Chengdu 611130, China

**Keywords:** carbon partitioning, leaf plasmodesmata, photosynthesis, sink, shade

## Abstract

The border row crop in strip intercropped maize is often exposed to heterogeneous light conditions, resulting in increased photosynthesis and yield decreased. Previous studies have focused on photosynthetic productivity, whereas carbon allocation could also be one of the major causes of decreased yield. However, carbon distribution remains unclear in partially shaded conditions. In the present study, we applied heterogeneous light conditions (T), and one side of plants was shaded (T-30%), keeping the other side fully exposed to light (T-100%), as compared to control plants that were exposed entirely to full-light (CK). Dry weight, carbon assimilation, ^13^C abundance, and transport tissue structure were analyzed to clarify the carbon distribution in partial shading of plants. T caused a marked decline in dry weight and harvest index (HI), whereas dry weight in unshaded and shaded leaves did not differ. Net photosynthesis rate (Pn), the activity of sucrose phosphate synthase enzymes (SPS), and sucrose concentration increased in unshaded leaves. Appropriately, 5.7% of the ^13^C from unshaded leaves was transferred to shaded leaves. Furthermore, plasmodesma density in the unshaded (T-100%) and shaded (T-30%) leaves in T was not significantly different but was lower than that of CK. Similarly, the vascular bundle total area of T was decreased. ^13^C transfer from unshaded leaves to ear in T was decreased by 18.0% compared with that in CK. Moreover, ^13^C and sucrose concentration of stem in T were higher than those in CK. Our results suggested that, under heterogeneous light, shaded leaves as a sink imported the carbohydrates from the unshaded leaves. Ear and shaded leaf competed for carbohydrates, and were not conducive to tissue structure of sucrose transport, resulting in a decrease in the carbon proportion in the ear, harvest index, and ear weight.

## 1. Introduction

Light plays an essential role in the regulation and development of carbon partitioning in plants [1,2,3] Sucrose is the major translocation form of photosynthetically-assimilated sugars in plants; its transportation often takes two pathways, namely, apoplastic and symplastic [4,5]. The apoplastic loading pathway and sucrose enters the phloem parenchyma cells from the leaf source cells through plasmodesmata. However, sucrose passes through the apoplast between the phloem parenchyma cells and the companion cells. Sucrose moves through a sucrose channel in the plasma membrane and is transported by a proton/sucrose co-transporter [6]. Symplast loading is a passive process, driven by a high concentration gradient of sucrose from the mesophyll to the phloem. This process does not require energy consumption but requires highly dense intercellular plasmodesmata [7,8,9]. In addition, the number of plasmodesmata in leaves is positively correlated with the net photosynthesis rate (Pn) in C4 grass [10]. However, light, as one of the factors affecting the anatomy of the loading zone, could also influence the plasmodesmata density [11,12]. In general, when a plant is exposed to high amounts of light, plasmodesmata were intensely developed, resulting in higher plasmodesmata density in contrast to the plants grown under low light conditions [13,14]. Subsequently, these numerous plasmodesmata were advantageous to sucrose transportation under high photosynthetically active radiation. Functionally, plasmodesmata at the bundle and vascular parenchyma (VP) cell interface are orchestrated and regulated through callose formation. In this way, sucrose is rapidly transported through a sympathetic gradient along with the cell-loading pathway from the Kranz mesophyll to a VP under high light [13]. However, in shaded leaves, sucrose is slowly transported along the gradient and accumulates in mesophyll cells, resulting in feedback inhibition of photosynthesis [15,16,17]. Moreover, large and small vascular bundles containing xylem and phloem are crucial components of sugar transportation. Any decrease in a vascular bundle under low light conditions results in a declined sucrose transportation capacity of leaves [18].

Previous reports have mainly focused on transportation and related tissue structure for carbon partitioning in plants, grown entirely under high or low light conditions. However, the overall local, as well as the individual environment of a plant leaf, manipulates its structural attributes, and the leaf adjusts its morphology accordingly. Eventually, the light environment of mature leaves alters the development of plants’ leaves area, thickness, palisade tissue, chloroplasts, and photosynthetic capacity [19,20,21]. Similarly, CO_2_ concentration surrounding mature leaves controls the stomatal density of developing leaves [22,23,24]. All these reports support the fact that systemic signals regulate leaf development. To the best of our knowledge, few people pay attention to the effect of leaf structure on carbon allocation under heterogeneous light conditions.

Another factor affecting carbon partitioning is the source–sink relationship. In general, carbon assimilates translocated from a source to sink; however, the source and sink of crops are dynamically changing [6]. Interestingly, developing leaves, instead of producing carbon assimilates, acts as a sink for carbon assimilates. Nevertheless, this input gradually decreases with leaf growth and as an output material begins to increase, such leaves gradually transform into the source of nutrient output [25]. However, leaf source is affected by environmental factors; leaf activity and carbon assimilation output decrease in low light, nitrogen deficiency, and the lack of water leading to a weak sink due to lack of nutrients [5,26,27]. In previous studies, mature leaves were regarded as the source and carbon assimilates output decreased under abiotic stress. However, whether mature leaves import carbon assimilates as a sink when abiotic stress (such as light) reaches a certain level remains unclear.

In practice, two or more crops grow in proximity in the same field during intercropping, forming a typical wide-narrow row in crop field planting [28,29,30]. The most common example of such cropping patterns is maize–soybean relay strip intercropping (MSR), which is being practiced worldwide [31,32,33]. Moreover, there is a growing interest of researchers to explore the heterogeneous light conditions (one side leaves in high light, and the other side leaves under low light) for the same plant in border-row crops in MSR [34,35]. As a result, regarding the maize in MSR, leaf Pn was decreased in leaves under low light (light intensity appropriately 30% less than natural light) and increased in leaves under high light conditions [35]. Interestingly, the whole plant in intercropping exhibits high photosynthesis but less yield compared with the monoculture [35]. Such a discrepancy indicates that maize leaves, despite the sufficient photosynthesis, may undergo carbon partitioning changes, and obstructed sucrose transportation. Therefore, we speculated that a systemic irradiance signal regulates the tissue structure of maize leaves for sucrose transportation and mature leaves under low light conditions are transformed into the sink to import carbon assimilates.

This study aimed to determine the response of the tissue structure for transportation to system signal regulation under heterogeneous light conditions and to reveal whether shaded leaves are acting as a sink, which results in a decrease in the carbon assimilation translocated to ear and decrease in yield. Hence, we analyzed the dry weight of organs, harvest index (HI), photosynthetic capacity, number and size of starch granules, sugar concentration, ^13^C abundance, the density of plasmodesmata, vascular bundle numbers and area of ear stem in maize plants grown in a different light environment. In one group of plants, leaves were under shade on the left side (T-30%) and under natural light on the right side (T-100%). In another group, all leaves were under full sunlight (CK).

## 2. Results 

### 2.1. Dry Weight and HI

Dry weight and HI were determined to evaluate the accumulation and partitioning of carbohydrates in plants. The average dry weight of leaves (T-30% and T-100% leaves) was 10.6% less than the dry weight of leaves in CK (average CK-left and CK-right). The dry weight of stems and ears in T treatment decreased by 26.4% and 50.2%, respectively, compared to those of CK (Figure 1A,B). The total dry weight and HI in T were significantly lowered by 39.2% and 18.0%, respectively, compared to those under CK (Figure 1C,D). However, the T-30% leaves dry weight was not considerably lower in comparison with that of T-100%, the CK-left, and CK-right.

### 2.2. Net Photosynthesis Rate

Pn in the T and CK groups was measured to examine further the capacity of carbon assimilation in shaded leaves and full sunlight leaves. Compared with T-30%, T-100% showed significantly higher Pn by 134.9% (Figure 2). Interestingly, T-100% showed significantly higher Pn compared with the CK-left and CK-right (although the leaves in the T-100%, CK-left, and CK-right groups were subjected to normal light). Pn values in CK-left and CK-right groups showed no difference. These results suggest that unshaded leaves in T showed better carbon assimilation capacity compared with CK leaves.

### 2.3. ^13^C Abundance

Ear leaves in the T-100% and CK-right groups were labeled with ^13^CO_2_ for one h to analyze ^13^C abundance in organs (Figure 2B–D). The ^13^C in the T-30% and T-100% leaves were 5.7% and 6.4%, respectively (showing no significant difference); however, ^13^C in the CK-left leaves was significantly lower than that in the CK-right leaves. Approximately 5.7% ^13^C in T-100% leaves exported to T1-30% leaves (in addition to the natural abundance of ^13^C, the ^13^C of T-30% leaves comes from T-100% leaves), the ^13^C of CK-right leaf transfers 2.1% to CK-left leaf (^13^C of T-left leaves comes from T-right), the transfer amount of T-100% was 270.0% of CK-right. ^13^C in the stems and ears of the T groups was significantly higher and lower than that in the CK, respectively. The shaded leaves imported carbon from unshaded leaves, which increased carbon accumulation in plant stem under the T treatment.

### 2.4. Starch Granules and Chloroplast Ultrastructure

The effects of partial shading treatment on the ultrastructure of the leaves were presented in Figure 3. The number of starch granules in the T group was 55.1% less than that in the CK groups, as shown in the TEM images. Furthermore, the amounts of starch granules were similar in T-30% and T-100% leaves (Figure 3A,B). Similarly, the number of starch granules in the CK-left and -right were also the same (Figure 3C,D). The size and number of starch granules per chloroplast and chloroplast per bundle sheath cell were statistically analyzed (Figure 3E,F). The T group (T-30% and T-100%) showed significantly less number of starch granules compared with the CK groups (CK-left and CK-right). Starch granules size of chloroplasts did not differ between the T and CK groups. The number of starch granules and size of starch granules and chloroplasts size did not differ between the T-30% and T-100% groups and between the CK-left and CK-right groups. However, T-30% showed higher chloroplast numbers compared with the T-100%.

### 2.5. Sugar Concentration 

Plants under T treatment exhibited lower starch concentration compared with the CK group (Figure 4A,B). Starch concentration did not significantly differ between the T-30% and T-100% groups and the CK-right and CK-left groups. Leaves sucrose concentration in the T-100% group did not differ from that in the CK group, the sucrose concentration in T-100% was 22.6% higher than in the T-30%. Starch concentration in the stems and grains was significantly lower in the T group than that in the CK by 24.2% and 31.0%, respectively. However, the sucrose concentration of grain in T was higher than that of the CK group by 22.7% (Figure 4C,D).

### 2.6. The Activity of Key Enzymes for Sucrose and Starch Synthesis 

We also analyzed the activities of ADP-glucose pyrophosphorylase (AGPase), sucrose phosphate synthase (SPS), and sucrose synthase (SS) in leaves (Figure 5A–C). AGPase and SS are enzymes crucial for starch synthesis; these enzymes activity in the T group (T-30% and T-100% leaves) were significantly decreased compared with CK (CK-left and CK-right). SPS is an enzyme crucial to sucrose synthesis. SPS activity was significantly decreased in the T-30% by 19.3% than that in the CK group (CK-left and CK-right), SPS was higher in the T-100% group by 17.4% than that in the T-30% group. AGPase activity did not significantly differ between the T-30% and T-100% groups and in the CK-left and CK-right groups.

### 2.7. Plasmodesmata Density and Vascular Bundle

For plants under the partial shading treatment, the plasmodesmata were significantly affected (Figure 6E). The plasmodesmata density of maize leaves under T treatment was significantly lower than that under CK, but within the two groups, it did not differ between the left and right side leaves. In addition, the number of large vascular bundles number and cross-sectional area of ear stem in the T group was lower than that of the CK group (Figure 7). However, the number of small vascular bundles and the cross-sectional area increased under T treatment. Furthermore, the T group showed a decreased total area of vascular bundles in the T group by 27.5% compared with the CK group (Figure 7D).

## 3. Discussion 

### 3.1. Effect of Transportation Tissue Structure on Carbon Partitioning 

This study demonstrated that plasmodesmata density was regulated by system signals, affecting carbon distribution. The unshaded leaves under T treatment (i.e., T-100%; kept in natural light) showed a significant decrease of leaf plasmodesmata density compared with the CK (all leaves in natural light), as shown in Figure 6. Leaf anatomy of C4 and C3 plants were regulated by systemic irradiance signals under heterogeneous light. Previous results suggested that signals triggered by changes in irradiance are transmitted from mature to developing leaves, which regulate leaf development [22,36,37]. Actually, the shading of lower mature leaves decreases the stomatal density and index of upper developing leaves [23,24]. Likewise, when sorghum grows under heterogeneous light, the thickness and mesophyll cells of developing leaves (under shaded) and mature leaves (under normal light) are not significantly different [38]. In the current study, the plasmodesma density in leaves under natural light was inhibited by location irradiance of shaded leaves. Interestingly, the plasmodesma density of unshaded leaves (T-100%) was regulated by systemic irradiance signals from shaded leaves (T-30%), reciprocally, systemic signals from unshaded leaves regulated the anatomy structure of shaded growing leaves [19]. The current research involved the shading of many leaves (half of the leaves of the plant), including developing and mature leaves. However, we focused on the mature leaves (being stable representative) rather than on developing leaves. Similarly, Larcher [39] suggested that the detection of irradiance signal by mature parts of the plant, i.e., a wholly unfurled and mature leaf ensures the most accurate measurement of irradiance around the plant, rather than by (or in addition to) a developing leaf growing in a less representative irradiance. Importantly, regarding the partial shading of the plant, we report that plasmodesma density decreased in unshaded leaves, and these regulation signals came from shaded leaves, which has not been previously reported.

The plasmodesmata are important microscopic channels in the sucrose loading of the symplast pathway, which is then transported in the phloem over a long-distance [8,9]. Phloem loading of sucrose in maize was not only related to the density of plasmodesmata but also limited by callose deposition at plasmodesmata in bundle sheath-vascular parenchyma interface [40]. In the present study, the amount of ^13^C transferred was not affected by plasmodesmata density. Although the density of plasmodesmata decreased under heterogeneous light, the amount of ^13^C transfer from unshaded leaves to shaded leaves increased (Figure 2 and Figure 6). Moreover, the cross-sectional area of the vascular bundle was closely related to the transportation capacity of sucrose, the product of volume flux, cross-sectional path area, and concentration of a transported assimilate determines bulk flow rate [18]. The area of the vascular bundle decreased in the T group (Figure 7). The ear dry weight and HI in T treatment were lower than those under CK whiles the ^13^C proportion of ear in T decreased (Figure 1 and Figure 2). Thus, a HI decrease in partial shading of leaves could be due to the decreased vascular bundle area, which limited the photosynthate transport to the ear. However, we studied the transportation of sucrose by the symplast pathway and whether sucrose was transported by the apoplastic pathway still remains a question which should be investigated further. Future studies could be designed to compare the sucrose transport pathway including sucrose transport-related proteins, e.g., sucrose transporters (SUTs) and SWEETs genes family.

### 3.2. Shaded Leaves as a Sink Effect of Carbon Partitioning 

Plant carbon transportation is often described as a relationship between organs that produce more carbon than they consume (acting as a source) and those that use more carbon than they produce (acting as a carbon sink) [6,41,42]. In the present study, the decreased ear dry weight may be due to the fact that shaded leaves also act as a sink. Pn in unshaded leaves was higher than shade leaves (Figure 2A). Furthermore, SPS activity and sucrose concentration increased (Figure 4B and Figure 5B). Theoretically, the dry weight of unshaded leaves should increase due to the high assimilation rate. Surprisingly, unshaded (T-100%) and shaded (T-30%) leaves actually showed no significant difference in weight (Figure 1A). ^13^C labeling clarified that carbon was distributed to shaded leaves (Figure 2B). Moreover, the number of starch granules and starch concentration decreased in unshaded leaves (T-100%) (Figure 3 and Figure 4A) due to the much of triosephosphate synthesized sucrose and then transported to shaded leaves. Starch and sucrose are synthesized from triosephosphate [43,44,45].

Furthermore, the carbon assimilation rate of unshaded leaves was regulated by systemic signals. Under heterogeneous light condition (T), Pn improved in the unshaded leaves (T-100%) as compared with the CK (Figure 2A). Leaf photosynthesis was determined by their local environment and by specific systemic signals from other parts of the same plant. The light-dependent systemic regulation of the chloroplast structure, photosynthesis pigments, quantum efficiency of photosystem Ⅱ (ΦPSⅡ), and rubisco, along with their respective effects on photosynthesis, have been studied in dicots and monocots previously [19,20,46,47]. The present study showed that the chloroplast number decreased in unshaded leaves compared with the shaded leaves, but Pn was higher than shaded leaves (Figure 3G). Theoretically, a low amount of chloroplast is possible, not conducive to carbon fixation, and the carbon assimilation rate would decreases. Similarly, some studies showed that, in maize, sorghum, and soybean, although the chloroplast structure does not benefit the carbon fixation, the changes in physiological indices, such as an increase in photosynthetic pigment, ΦPSⅡ, electron transport rate, and rubisco content, resulted in the improvement of leaf Pn under heterogeneous light conditions [20,36,48]. Therefore, in this study, changes in photosynthetic physiological indexes may be the main reason for the increased Pn in unshaded leaves. Recently, Ehonen et al. [49] found that the carbon assimilation rate of distal leaves is related to the stomatal conductance regulated by system signals. In hybrid aspen and silver birch, leaves subjected to darkness treatment show Pn changes that are related to stomatal conductance in leaves under normal light. The systemic signal is perceived directly by guard cells, leading to a decrease in stomatal conductance followed by a decrease in carbon assimilation rate due to a reduction in the supply of CO_2_ entering through the stomata. Future research should investigate the single systemic regulation of stomatal conductance and its influence on Pn.

## 4. Materials and Methods

### 4.1. Experimental Design and Plant Growth 

The experiment was conducted in 2019 at the Sichuan Agricultural University, Chengdu (30°42′ N 103°51′ E) in Sichuan Province, China. Maize was subjected to total radiation of 1496 MJ·m^−2^ during the growth period in 2019. The soil properties were as follows: pH, 6.73 (1:2.5, soil: water); Olsen-P, 3.0 mg·kg^−1^; organic matter, 8.3 g·kg^−1^; total N, 0.21 g·kg^−1^; available K, 103 mg·kg^−1^; and available N, 100 mg·kg^−1^.

The maize variety Zhongyu 3 was used in this study. It was a pot experiment, and pots were buried 30 cm below the ground. Maize seeds were sown on 28 March 2019, the area per plant was 4.5 m^−2^. The maize plants were exposed to heterogeneous light conditions and compared with the plants under normal light conditions on both sides (CK). The right-side leaves (from top to bottom) of the treatment plants (T) were exposed to 30% of a normal light (T-30%, photosynthetically active radiation 421.1 μmol·m^−2^·s^−1^) while all left-side leaves from top to bottom were in normal light (T-100%, photosynthetically active radiation 1477.5 μmol·m^−2^·s^−1^) (Figure 8). Light transmittance was controlled by a shading cloth with different porosities. Shading was initiated at stage V6 (sixth-leaf) and ended at physiological maturity (134 days after sowing, a layer of black cells develops at the kernel base). The plants were harvested on 12 August.

During the growth period of maize, 18.9 g urea (with N 46%), 10 g superphosphate (with P_2_O_5_ 12%), and 2.5 g potassium chloride (with K_2_O 60%) were applied to each plant. Each treatment had three replicates, i.e., three times sampling (plants with ear leaf in unshaded were used to measure the data), with three extra plants, and 24 plants for each treatment.

### 4.2. Measurement of Dry Weight and Harvest Index 

After sowing for 134 days, stem, left-leaves, right-leaves, and ears of three plants were sub packaged and dried at 80 °C to a constant weight. HI was calculated as follows: HI = dry weight of grain/total dry weight.

### 4.3. Measurement of Pn 

Pn was measured by a photosynthesis system (Li-6400XT) equipped with 6400-02B red/blue LED light source from 10:00 AM to 11:30 AM at 15 days after silking (DAS). The control conditions were manually set to a CO_2_ concentration of 400 μmol CO_2_ mol^−1^, light intensity 1000 μmol m^−2^ s^−1^, sample cell 25 °C, leaf temperature (24–26 °C), air temperature approximately (25–28 °C), and relative humidity of appropriately (65–70%). Three plants were selected to determine Pn of ear leaf and superior leaf.

### 4.4. Measurement of ^13^C Abundance

Based on the method of Liu [50] with modifications, three plants were chosen with ear leaf under full sunlight, polyethylene plastic bags (98% transmittance) with a width of 15 cm and a length of 80 cm were placed on unshaded ear leaves (CK coated on the ear leaves) at 15 DAS (sunny and cloudless weather). We sealed the mouth of the plastic bags with double-sided foam tape, the bags were filled (with a syringe) with 60 mL of ^13^CO_2_ (99.9% purity) and then removed after 1 h. Before removing the bags, a syringe was used to suck out the remaining CO_2_, and then slowly injected to the NaOH solution for recycling. After 24 h, the stems, ears, left-leaves, and right-leaves were dried to a constant weight at 80 °C. The ^13^C abundance of each part was determined using an elemental analyzer (E.A.; Flash2000, Thermo Fisher Scientific, MA, USA) and isotope ratio mass spectrometer (Delta V IRMS Thermo Fisher Scientific, MA, USA) [51].

### 4.5. Observation of Starch Granules, Chloroplasts, Intercellular Plasmodesmata, and Vascular Bundles 

Three ear leaf and superior leaf samples were obtained at 15 DAS, pre-fixed with a solution of 3% glutaraldehyde, post-fixed in 1% osmium tetroxide, dehydrated in a series of acetone, infiltrated in Epox 812, and then embedded. The semi-thin sections were stained with methylene blue. Ultrathin sections were cut with a diamond knife and then stained with uranyl acetate and lead citrate. Sections were examined with a transmission electron microscope (TEM; HITACHI, H-600IV, TYO, Japan) [52]. Images were captured and plasmodesmata density was determined on the basis of numbers of plasmodesmata per 5 μm. The number of chloroplasts and starch granules (choosing 20 chloroplasts and counting the number of starch granules in each chloroplast) was measured by the ImageJ software (NIH, MD, USA). Ear stem on full sunlight, the third node of the ear stem (15 DAS) was cute, embedded in paraffin, sliced, dewaxed, and photographed by fluorescence microscopy. ImageJ was used for measurements of the number and area of vascular bundles.

### 4.6. Measurement of Sugar and Enzyme Activity 

Three maize plants with similar growth were selected to determine sucrose and starch concentrations in the ear leaf, superior leaf stems and grains at 15 DAS. Sucrose was measured directly in the extract with resorcinol as the coloring reagent (Shi et al., 2016). Starch was determined via acid hydrolyzation [53]. The activities of sucrose phosphate synthase (SPS), sucrose synthase (SS), and ADP-glucose pyrophosphorylase (AGPase) in-ear leaf and superior leaf were measured as previously described [54,55].

### 4.7. Statistical Analyses 

Different treatments were analyzed using one-way ANOVA in the SPSS statistical software package (Version 19.0, IBM SPSS Statistics, NY, U.S.). Significant differences were assessed by Least Significant Difference (LSD) at *p* < 0.05. Origin software (Version 8.0, OriginLab, MA, USA) was used for mapping.

## 5. Conclusions

This study provides new insights into the systemic regulation of leaf structure. For maize grown under a heterogeneous light environment, the plasmodesmata density of unshaded leaf decreased, which was regulated by systemic irradiance signals from shaded leaves. Furthermore, the isotopic tracing of photoassimilate in maize reveals that the shaded leaves act as a sink during the early grain filling phase of phase, and they utilized most of the photoassimilates developed by unshaded leaves. Moreover, our results showed that, during the early grain filling stage, plasmodesmata played a weak role in the assimilate allocation of the source to sink, while the cross-sectional area of the vascular bundle was reduced, which hinders the translocation of photoassimilates to developing seeds in-ear. Thus, shaded leaves and ears competed for assimilating and unfavorable structure of vascular bundle, which decreases the final yield of maize plants.

## Figures and Tables

**Figure 1 plants-09-00987-f001:**
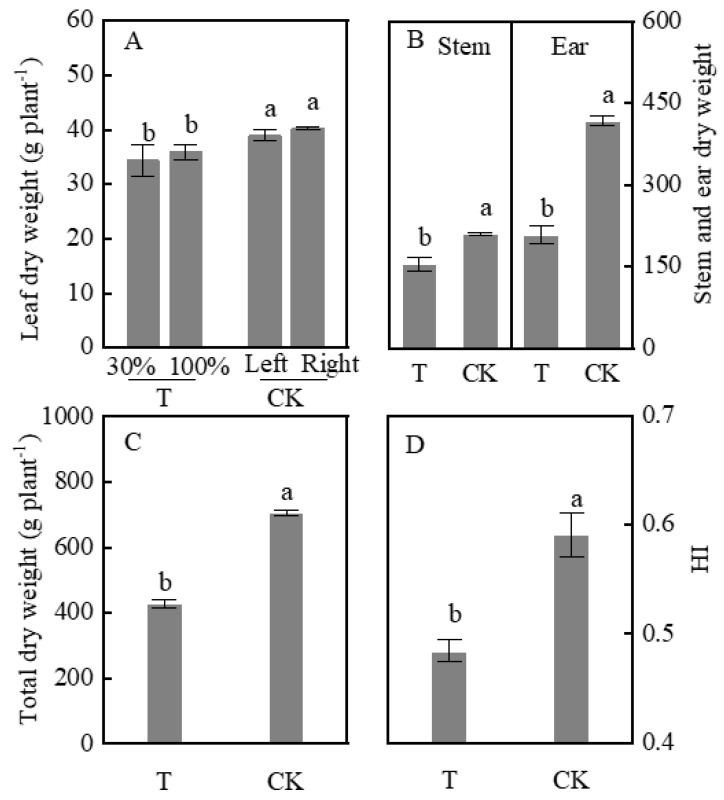
Leaf, stem, total dry matter weight and harvest indices (HI) in shading treatments. (**A**): leaf dry weight; (**B**): stem and ear dry weight; (**C**): total dry weight; (**D**): HI. Data are averages of three replicates, and bars represent standard errors. Plants grown under heterogeneous light (T), the left side leaves were under shade (T-30%) and right side leaf under natural light (T-100%). All leaves were under full sunlight (CK). Data with different letters are significantly different (*p* < 0.05).

**Figure 2 plants-09-00987-f002:**
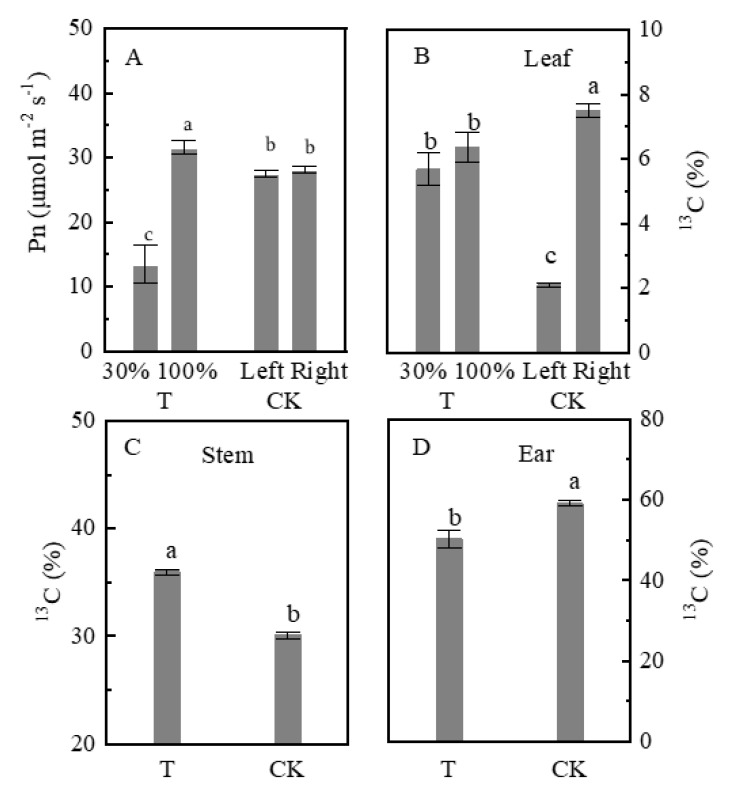
Net photosynthetic rate (Pn) in leaves and ^13^C abundance of the organ in shading treatments. (**A**): leaf Pn; (**B**): ^13^C abundance of leaf; (**C**): ^13^C abundance of stem; (**D**): ^13^C abundance of ear. Ear leaves were labeled with ^13^CO_2_ in the T-100% and CK-right groups. Plants grown under heterogeneous light (T), the left side leaves were under shade (T-30%) and right side leaf under natural light (T-100%). All leaves were under full sunlight (CK). Data are averages of three replicates and bars represent standard errors. Data with different letters are significantly different (*p* < 0.05).

**Figure 3 plants-09-00987-f003:**
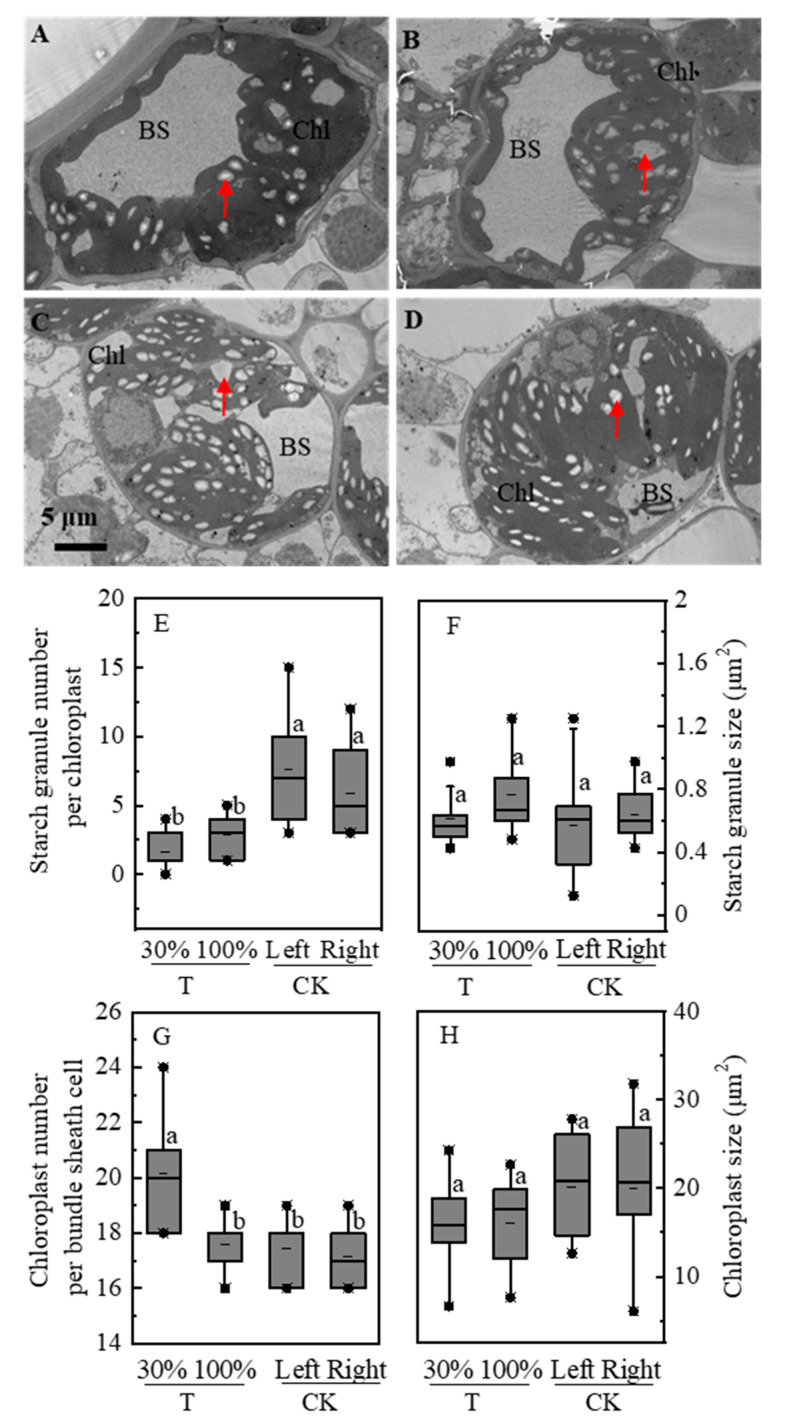
Starch granules, chloroplast number, and size of leaves in shading treatments. The arrow points to starch granules, Chl: chloroplasts, BS: bundle sheath cells. (**A**): starch granules and chloroplasts of T-30%; (**B**): starch granules and chloroplasts of T-100%, the ear leaf in full sunlight; (**C**): starch granules and chloroplasts of CK-left; (**D**): starch granules and chloroplasts of CK-right; (**E**): starch granule number per chloroplast; (**F**): starch granule size; (**G**): chloroplast number per bundle sheath cell; (**H**): chloroplast size. The short solid line within each box represents the median and mean values of all data. Plants grown under heterogeneous light (T), the left side leaves were under shade (T-30%) and right side leaf under natural light (T-100%). All leaves were under full sunlight (CK). Data with different letters are significantly different (*p* < 0.05).

**Figure 4 plants-09-00987-f004:**
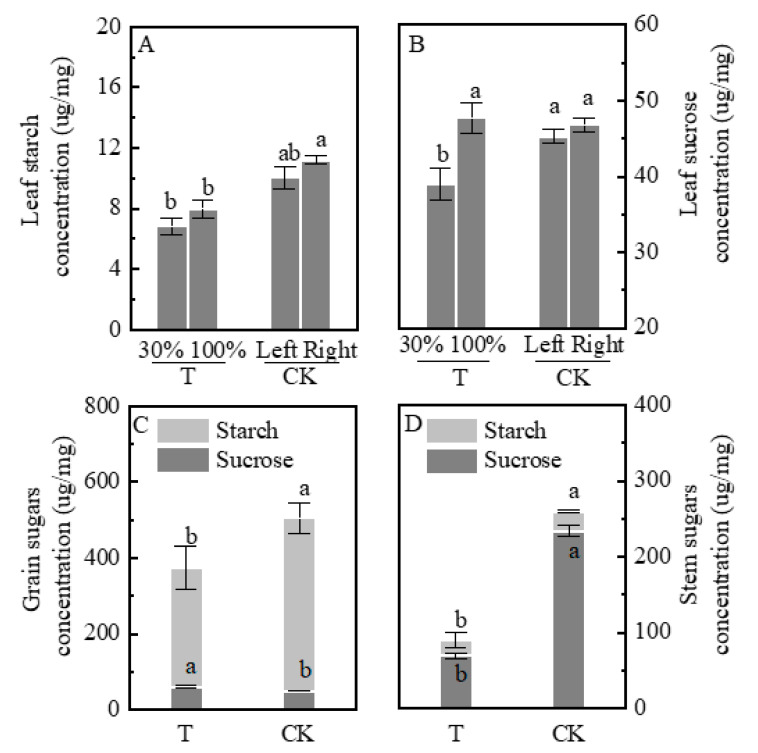
Starch and sucrose concentrations in organs in shading treatments. (**A**): leaf starch concentration; (**B**): leaf sucrose concentration; (**C**): grain starch and sucrose concentration; (**D**): stem starch and sucrose concentration. Plants grown under heterogeneous light (T), the left side leaves were under shade (T-30%) and right side leaf under natural light (T-100%). All leaves were under full sunlight (CK). Data are averages of three replicates and bars represent standard errors. Data with different letters are significantly different (*p* < 0.05).

**Figure 5 plants-09-00987-f005:**
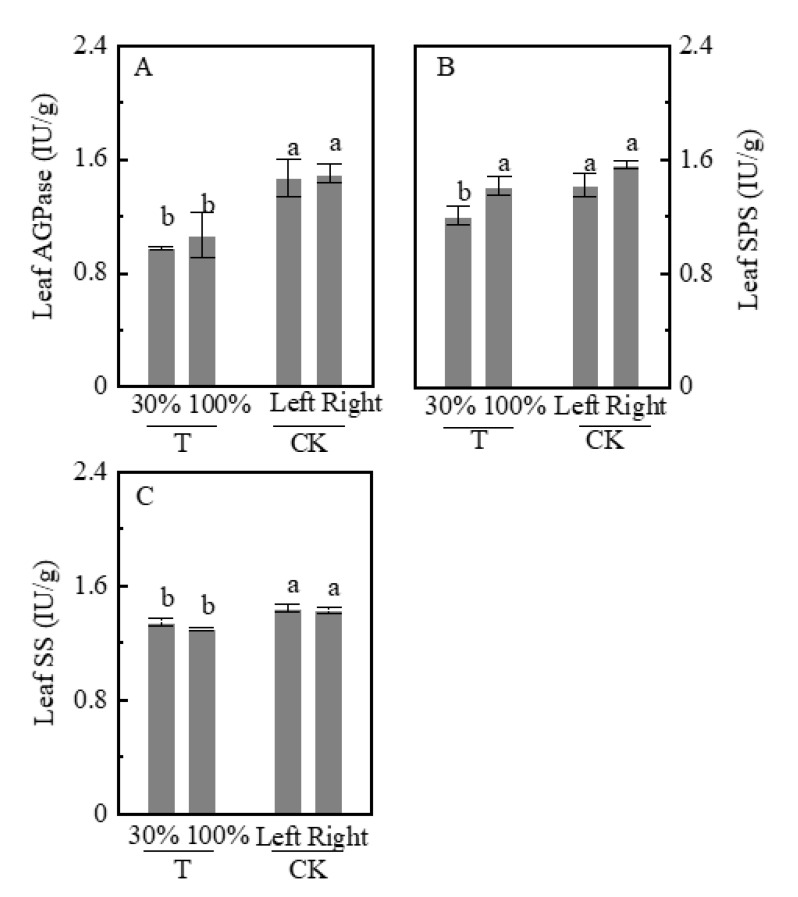
Activities key enzymes in sucrose and starch synthesis in leaves. (**A**): ADP-glucose pyrophosphorylase (AGPase); (**B**): sucrose phosphate synthase (SPS); (**C**): sucrose synthase (SS); IU, 1 IU is the amount of enzyme that catalyzes the reaction of 1 μmol of substrate per minute. Data are averages of three replicates and bars represent standard errors. Plants grown under heterogeneous light (T), the left side leaves were under shade (T-30%) and right side leaf under natural light (T-100%). All leaves were under full sunlight (CK). Data with different letters are significantly different (*p* < 0.05).

**Figure 6 plants-09-00987-f006:**
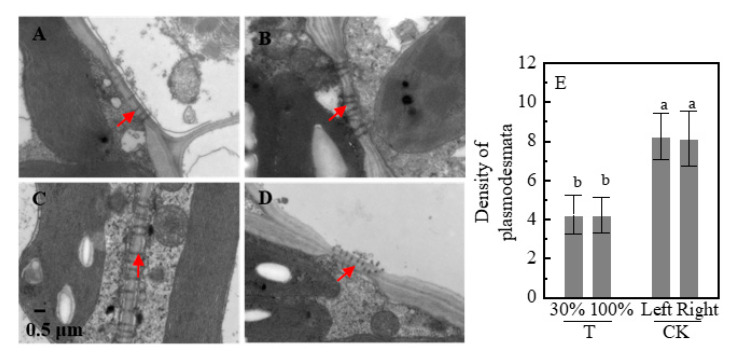
Plasmodesma density of vascular bundle sheath cells and vascular parenchyma cells of the leaves in shading treatments. Scale bar, 0.5 μm. The arrows point to plasmodesma. Plasmodesma density is the number of plasmodesma within five μm of the vein. (**A**): T-30%; (**B**): T-100%; (**C**): CK-left; (**D**): CK-right; (**E**): density plasmodesmata. Plants grown under heterogeneous light (T), the left side leaves were under shade (T-30%) and right side leaf under natural light (T-100%). All leaves were under full sunlight (CK). Data with different letters are significantly different (*p* < 0.05).

**Figure 7 plants-09-00987-f007:**
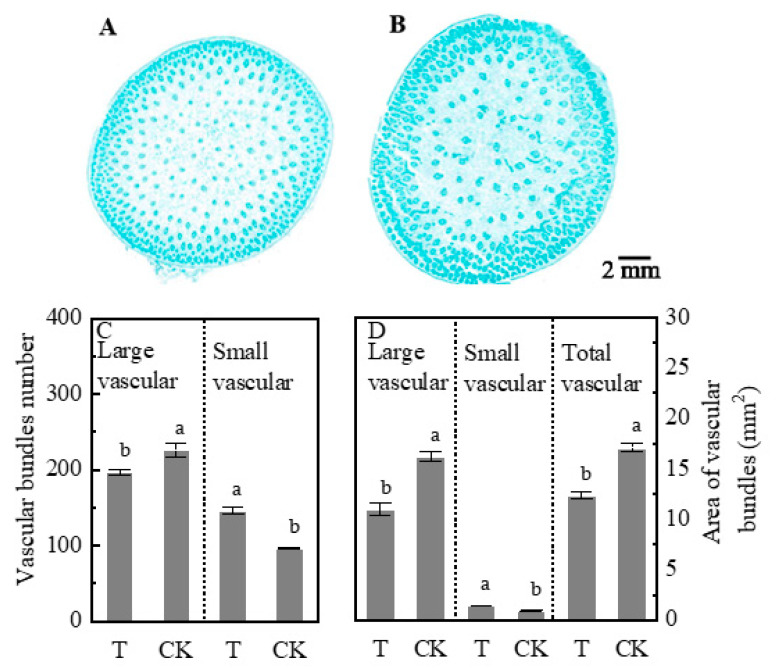
The number of large and small vascular bundles. The cross-sectional area of large and small vascular bundles and the number of vascular bundles in an ear stem. Small vascular bundles are numerous, small, and closely arranged in the peripheral portion. Large vascular bundles, towards the center, the bundles are comparatively large in size and loosely arranged. (**A**): T treatment; (**B**): CK. Scale bar, 2 mm, ear stem in full sunlight; (**C**): vascular bundles number; (**D**): area of vascular bundles. Plants grown under heterogeneous light (T). All leaves were under full sunlight (CK). Data with different letters are significantly different (*p* < 0.05).

**Figure 8 plants-09-00987-f008:**
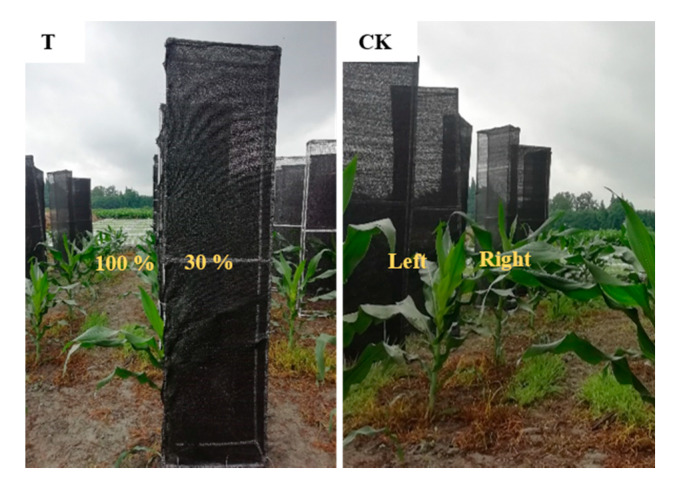
Design of shading treatments. T, all left-side leaves in maize plants from top to bottom were under normal light, whereas all the right-side leaves from top to bottom were in 30% of normal light. CK, the left and right maize leaves were under normal light.

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
