# Peer review of "Heterogeneous Light Conditions Reduce the Assimilate Translocation Towards Maize Ears"

_plants, 2020, doi:10.3390/plants9080987_

Round 1

Reviewer 1 Report

"The shade leaves not the ear as a sink at early filling stage, decreased the final maize yield under heterogeneous light condition"

The manuscript is aimed to determine whether maize transport tissue structure is regulated by systemic irradiance signal and if shaded leaves acting as a sink that influenced carbon partitioning.

The investigation is very interesting, giving the new insight into the functioning of shaded leaves that act as a sink and not the ear as previously thought. The hypothesis is clearly stated and the authors used considerable number of methods what covered wide range of parameters to prove their hypothesis.

However, I believe that most part of the manuscript should be rewritten to support their hypothesis and to be easier to read.

There are some additional points that authors have to consider:

The authors should consider changing the title so it would be more attractive. In this form is rather confusing.

The results are confusing and it should be written systematically so it will be easier to read.

The discussion lack in-depth analysis and should be supported by some additional citations that refers to C4 plants or at least to monocots, not only soybean that is dicot.

The authors should consider using some help for English language editing of style and grammar.

In the pdf file, there are additional comments that authors should take into account to correct and improve their manuscript.

Best regards.

Author Response

Dear Reviewer,

Thank you for considering our article for publication in Plants. We would like to thank you for a careful and thorough reading of this manuscript and for the thoughtful comments and constructive suggestions, which have helped to improve the quality of this manuscript. We have discussed and considered earnestly about them and revised our manuscript accordingly. The following are a list of our responses to the reviewers’ comments.

Kind regards,

Xiaochun Wang

Below are our responses to you and the reviewers.

Reviewer #1

General comments

  1. Comments: The results are confusing and it should be written systematically so it will be easier to read.

Response: As suggested by the reviewer, we re-write the results for a clear understanding of readers. Please see the result section in the revised manuscript. 

  1. Comments: The discussion lack in-depth analysis and should be supported by some additional citations that refers to C4 plants or at least to monocots, not only soybean that is dicot.

Response: As suggested by the reviewer, we have added some references related to C4 plants, and revised the discussion. Please see it in the discussion section.

  1. Comments: The authors should consider using some help for English language editing of style and grammar.

Response: As suggested by the reviewer, we have invited a good English editor (Professor from our department) to refine the grammar and language.

Detailed Comments:

  1. Comment: The title is confusing and it should be changed.

Response: As requested by the reviewer, the title changed to “Heterogeneous light conditions reduce the assimilate translocation towards maize ears”.

  1. Comment: Please correct affiliation.

Response: We have corrected and added the affiliation on lines 8 and 11.

  1. Comment: This sentence is incomprehensible. Please correct!

Response: We have revised the sentence and corrected our mistakes. Please see lines 15, 16, and 31 in the revised manuscript.

  1. Comment: There should be one or two sentences about apoplastic pathway.

Response: As requested by the reviewer, two sentences about the apoplastic pathway have been added to lines 40 to 44 in the revised manuscript.

  1. Comment: Of what? ‘high concentration gradient’

Response: Sir, this is a high concentration gradient of sucrose. We have corrected this error within the line 45.

  1. Comment: Use different word please! “Collectively”

Response: Sir, we have changed this word to “All” in line 67.

  1. Comment: Please rephrase this sentence (line 66-67), it is confusing.

Response: Sir, the correction has been done. Please see lines 67-69 in the revised manuscript.

  1. Comment: Please rephrase this sentence (line 69 and 70), it is incomprehensible.

Response: Sir, we have corrected this sentence. Lines 70-72.

  1. Comment: “the heterogeneous” extra space

Response: As suggested by the reviewer, we deleted the extra space.

  1. Comment: Is this your result? Please correct the grammar, it should not be in present tense.

Response: Thank you for your careful reading and our correction, Sir, we revised the sentences. Lines 85-87.

  1. Comment: I feel like the aim should be place before the hypothesis and it should be rewritten to be more clearly described what was the aim.

Response: As suggested by the reviewer, we have moved this aim at the start of the paragraph, and revised the aim. Lines 93-96.

  1. Comment: How did you calculated this % since you measured dry weight in T-30 and T-100 and in left and right part f the plant. I believe that it should be written in more details.

Response: As suggested by the reviewer, we have added these details in materials and methods in lines 104-105.

  1. Comment: ...compared to...

Response: We changed the “related” to “compared”. Line 106.

  1. Comment: ... higher...

Response: As suggested by the reviewer, we changed the “increased ” to “higher”. Line 118.

  1. Comment: “to analyse” extra space

Response: We removed the extra space.

  1. Comment: This should be rewritten, it is confusing! Where can be seen this results, I am having trouble to find it.

Response: We have rewritten the said lines, please see the lines 128-129.

  1. Comment: The figures should be of better quality, they are blured...

Response: As suggested by the reviewer, we have changed all blured figures to high quality figures. Please see the figures in the revised manuscript.

  1. Comment: How much is less?

Response: Dear Sir, we added the percentage of differences; please see line 140 in the revised manuscript.

  1. Comment: How many chloroplast did you use for counting the amount of starch granules? Please explain this in more details (in M&M section).

Response: Sir, we calculating starch granules from 20 chloroplasts and added all the details in materials. Please see the lines 333 and 334 in the revised manuscript.

  1. Comment: What part of the plant is ear left, it is confusing. You should explain it in more details.

Response: Sir, we have explained this in lines 153 and 154, 335.

  1. Comment: “Activity of key enzymes for sucrose and starch synthesis” This section is very confusing, it should be rewritten.

Response: Sir, we have revised this part and corrected our mistakes. Please see lines 170-175 in the revised manuscript.

  1. Comment: “critical” it should be crucial

Response: As suggested by the reviewer, we changed this.

  1. Comment: Both sides of the plant or?

Response: As required by the reviewer, we have added details in lines 172,173 and 175 in the revised manuscript. 

  1. Comment: Why likewise? It should be removed or explained better what is likewise.

Response: As suggested by the reviewer, we removed the word likewise.

  1. Comment: There is something wrong in this picture, i think you are missing the letter b, please check this and correct.

Response: As suggested by the reviewer, we have corrected the mistake within the picture. Please see (Figures 5) in the revised manuscript.

  1. Comment: Please, add the explanation of IU in the figure captions.

Response: As suggested by the reviewer, we explained IU in the figure caption. Lines 180-181.

  1. Comment: Were the T stems on full sunlight or shaded? Please explanin!

Response: As requested by the reviewer, the ear stems on full sunlight. We have added this detail in line 203.

  1. Comment: From which part of the stem were taken the samples for cross section of the stems? This should be explained in more details in M&M section.

Response: Sir, the third node of the ear stem was sampled and added the details in the M&M section. Line 335.

  1. Comment: This sentence is too long and confusing, please rewrite.

Response: Sir, thank you for your concerns. We revised the said lines. Lines 213-215.

  1. Comment: Please move this citation in line 214, behind the name of the author.

Response: As suggested by the reviewer, we moved this citation to line 222.

  1. Comment: “participant” Please use different word.

Response: Yes, Sir, we revised the words. Line 228.

  1. Comment: Please rewrite this part (lines 240-244), it is confusing.

Response: As suggested by the reviewer, we revised this part (lines 254-256).

  1. Comment: Concentration of what?

Response: Sir, it was the “starch concentration,” and we corrected the mistake.

  1. Comment: Please, ad ...in dicots and monocots... after the word previously, since references are referred to soybean and sorghum.

Response: As suggested by the reviewer, we have added “in dicots and monocots” to lines 262 and 263.

  1. Comment: please correct ...but Pn was higher...

Response: Sir, we have corrected the “ but Pn higher” to “ but Pn was higher”. Please see line 264 in the revised manuscript.

  1. Comment: Please move this citation in line 259, behind the name of the author.

Response: As suggested by the reviewer, we have moved the citation to line 271.

  1. Comment: This part of the M&M should be rewritten. It should be stated more clearly when was the experiment conducted. When the samples were taken, in how many replicates? Not all the readers of this Journal know the maturity stages of the maize and it should be explained in.

Response: As suggested by the reviewer, we have rewritten the part of the M&M and added more details, as you directed us to include.

  1. Comment: Did you measured the light intensity? Please add this information.

Response: As suggested by the reviewer, we have added information on light intensity. Lines 290-292.

  1. Comment: How can you measure this? When this physiological maturity occured? please explain.

Response: As suggested by the reviewer, physiological maturity: 134 days after sowing, a layer of black cells develops at the kernel base. We added this details on lines 294-295.

  1. Comment: This sentence (line 298-299) is confusing, prelase rewrite.

Response: As suggested by the reviewer, we have rewritten this sentence, please see line 322.

  1. Comment: This part of M&M should be rewritten in more details. Please see comments in Results section. Also, in the the last sentence you should write that you used ImageJ for measurements.

Response: As suggested by the reviewer, we have added more details. In the last sentence. Please see lines 327 and 333-337 in the revised manuscript. 

  1. Comment: What does this mean? I don`t understand, please rewrite in more details.

Response: As suggested by the reviewer, this sentence is redundant and has been deleted. Please see the line 347.

  1. Comment: This part is the highlight of this work, it is confusing and it should be rewritten so it would be easier to understand ant to read it.

Response: As suggested by the reviewer, we wrote a new conclusion for our manuscript. Lines 351-360.

Once again, we appreciate all the comments and suggestions from you and the reviewers. We learned a lot from your comments. Sir, I am not a native speaker; however, I tried my best to write in a good way. Thank you for your comments and suggestions.

Reviewer 2 Report

Using maize plants grown in the field conditions, Chen et al. performed a partial shading experiment and found that shaded leaves received a significant amount of carbon translocated from the unshaded ear leaves with an 13C labelled CO2 experiment. They also demonstrated that a plasmodesmatal density of the bundle sheath cells in the unshaded leaves could be regulated by a systemic signaling. The manuscript is overall well written and the data are interesting. On the other hand, the description of the methods is unclear to make the work so reliable. I will recommend that the manuscript is revised based on my following comments.

  1. Figure 1 B shows that the ear weight reduced in the T treatment. Were ears grown under “unshaded” conditions also for the T treatments? Which at leaf position does ear emerge in maize plants?
  2. Figure 7: Please explain clearly how the small and large vascular bundles were discriminated.
  3. Line 277: “ended at physiological maturity” is a vague term. The authors need to explain how many days the treatments were continued for.
  4. Line 289-: The light intensity for Pn measurements is not described. The authors need to assure that the light intensity and other conditions such as a leaf temperature were not different between the treatments, although there was a significant difference in Figure 2A.
  5. Line 292: “400 umol mL-1” might be wrong. “400 umol mol-1 or 400 ppm” would be correct. Please see the manual.
  6. Line 296: The authors describe that the bags were placed on the ear leaves. I could not figure out how the authors coped with a CO2 leak issue from the plastic bag. Please explain more in detail.
  7. Line 298: The authors mention that the plants were harvested 12 h after 13C treatment. Does this meant that they were harvested at night? It would be important to note night or day for the sampling period because the translocation mainly occurs during the night.
  8. Lines 318-319: “Hashida et al. 2016” and “Zhang etal. 2012” are not found in the reference list.
  9. Line 449: “Terashima, Y.a”?
  10. Figures 1, 4 and 5: The term “Leaves” in the labels of the y-axes should be changed to “Leaf”.
  11. Line 271: “in his study” should be “in this study”.

Author Response

Dear Reviewer,

Thank you for considering our article for publication in Plants. We would like to thank you for a careful and thorough reading of this manuscript and for the thoughtful comments and constructive suggestions, which have helped to improve the quality of this manuscript. We have discussed and considered earnestly about them and revised our manuscript accordingly. The following are a list of our responses to you comments.

Kind regards,

Xiaochun Wang

Below are our responses to you and the reviewers.

Reviewer #2

Comments:

  1. Comment: Figure 1 B shows that the ear weight reduced in the T treatment. Were ears grown under “unshaded” conditions also for the T treatments? Which at leaf position does ear emerge in maize plants?

Response: Dear Sir, some ear developed under unshaded conditions, other ears developed under shaded conditions in T treatment. When we applied the shade treatments at that time, there were no ears on the maize stems. Whereas, plants with ear leaf under unshaded was used to measurement data, observation the tissue structure. The ear emerged on the 9th leaf from bottom to top. We added more details on line 298. Please see the revised manuscript.

  1. Comment. Figure 7: Please explain clearly how the small and large vascular bundles were discriminated.

Response: Sir, small vascular bundles, were numerous, small and tightly arranged in the peripheral portion. Large vascular bundles, towards the center, the bundles are comparatively large and loosely arranged. We have explained it. Please see the revised manuscript lines 200-202.

  1. Comment: Line 277: “ended at physiological maturity” is a vague term. The authors need to explain how many days the treatments were continued for.

Response: Sir, in this study, maize took 134 days after sowing to mature. We have added this information in lines 294 and 295.

  1. Comment: Line 289: The light intensity for Pn measurements is not described. The authors need to assure that the light intensity and other conditions such as a leaf temperature were not different between the treatments, although there was a significant difference in Figure 2A.

Response: As suggested by the reviewer, we have described these details related to light intensity and leaf temperature in lines 311 and 313.

  1. Comment: Line 292: “400 umol mL-1” might be wrong. “400 umol mol-1 or 400 ppm” would be correct. Please see the manual.

Response: As requested by the reviewer, we have corrected to 400 μmol CO2 mol-1. Please see line 311 in the revised manuscript.

  1. Comment: Line 296: The authors describe that the bags were placed on the ear leaves. I could not figure out how the authors coped with a CO2 leak issue from the plastic bag. Please explain more in detail.

Response: As suggested by the reviewer, seal the mouth of the plastic bags with double-sided foam tape, the bags were filled (with a syringe) with 60 mL of 13CO2 (99.9% purity) and then removed after one h. Before removing the bags, use a syringe to suck out the remaining CO2, and then slowly injected into the NaOH solution for recycling. Please see the picture below, and more details in lines 319-322.

  1. Comment: Line 298: The authors mention that the plants were harvested 12 h after 13C treatment. Does this meant that they were harvested at night? It would be important to note night or day for the sampling period because the translocation mainly occurs during the night.

Response: We made a writing error, plants harvesting 24 h after 13C treatment. We have corrected it. Please see the line 322 in the revised manuscript.      

  1. Comment: “Hashida et al. 2016” and “Zhang etal. 2012” are not found in the reference list.

Response: As suggested by the reviewer, we have added the two references to the reference list in lines 544 and 548.

  1. Comment: “Terashima, Y.a”?

Response: As suggested by the reviewer, we have been corrected the name in line 503.

  1. Comment: Figures 1, 4 and 5: The term “Leaves” in the labels of the y-axes should be changed to “Leaf”.

Response: As suggested by the reviewer, the term “Leaves” in Figures 1, 4 and 5 were changed to “Leaf.” Please see Figures 1, 4 and 5.

  1. Comment: Line 271: “in his study” should be “in this study.”

Response: As suggested by the reviewer, we have been changed the “his” to “this” on line 294 in the revised manuscript.

Once again, we appreciate all the comments and suggestions from you and the reviewers. We learned a lot from your comments. Sir, I am not a native speaker; however, I tried my best to write in a good way. Thank you for your comments and suggestions.

Round 2

Reviewer 1 Report

Dear authors,

thank you for taking suggestions into account.

Sincirely